# Rhamnolipid Nano-Micelles Inhibit SARS-CoV-2 Infection and Have No Dermal or Eye Toxic Effects in Rabbits

**DOI:** 10.3390/antibiotics11111556

**Published:** 2022-11-04

**Authors:** Alaa M. Ali, Harriet J. Hill, Gehad E. Elkhouly, Marwa Reda Bakkar, Nermeen R. Raya, Zania Stamataki, Yasmin Abo-zeid

**Affiliations:** 1Department of Pathology, Faculty of Veterinary Medicine, Cairo University, Giza 12211, Egypt; 2Institute of Immunology and Immunotherapy, University of Birmingham, Birmingham B15 2TT, UK; 3Department of Pharmaceutics and Industrial Pharmacy, Faculty of Pharmacy, Helwan University, Cairo 11795, Egypt; 4Helwan Nanotechnology Center, Helwan University, Cairo 11792, Egypt; 5Botany and Microbiology Department, Faculty of Science, Helwan University, Cairo 11795, Egypt

**Keywords:** SARS-CoV-2, rhamnolipids nano-micelles, hospital acquired infections, hand sanitation

## Abstract

Hand hygiene is considered to be the key factor in controlling and preventing infection, either in hospital care settings or in the community. Alcohol-based hand sanitizers are commonly used due to their rapid action and broad spectrum of microbicidal activity, offering protection against bacteria and viruses. However, their frequent administration during COVID-19 pandemic was associated with serious hazards, such as skin toxicity, including irritation, skin dermatitis, skin dryness or cracking, along with peeling redness or itching, with the higher possibility of getting infections. Thus, there is a need to find alternative and novel approaches for hand sanitation. In our previous publications, we reported that rhamnolipids nano-micelles had a comparable antibacterial activity to alcohol-based hand sanitizer and a lower cytotoxicity against human dermal fibroblast cells. In the current study, we investigated the antiviral activity of rhamnolipids nano-micelles against SARS-CoV-2. There was no cytotoxic effect on Vero cells noted at the tested concentrations of rhamnolipids nano-micelles. The rhamnolipids nano-micelles solution at 20, 78, and 312 µg/mL all demonstrated a significant (*p* < 0.05) decrease of virus infectivity compared to the virus only and the blank vehicle sample. In addition, an acute irritation test was performed on rabbits to further ascertain the biosafety of rhamnolipids nano-micelles. In the eye and skin irritation tests, no degree of irritation was recorded after topical application of rhamnolipids nano-micelles. In addition, histopathological, biomarker, and hematological analyses from animals treated with rhamnolipids nano-micelles were identical to those recorded for untreated animal. From the above, we can conclude that rhamnolipids nano-micelles are a good candidate to be used as a hand sanitizer instead of alcohol-based hand sanitizers. However, they must still be tested in the future among healthcare workers (HCW) in a health care setting to ascertain their antimicrobial efficacy and safety compared to alcohol-based hand sanitizers.

## 1. Introduction

Hospital-acquired infections (HAIs) are considered to be a major cause of morbidity and mortality, and are the second most prevalent cause of death, globally [1]. Various multicentric studies presented that around 3.5 to 12% of hospitalized patients acquired at least one HAIs [2,3], with a possibility of 10 million deaths by 2050 due to HAIs [4]. The emergence and re-emergence of micro-organisms, including viruses, pose a serious threat to human health at a global level [5]. This was exemplified by the COVID-19 pandemic, due to SARS-CoV-2 and its variants, which resulted in a global death toll of 6,515,964 persons, as reported by the John Hopkin dashboard on 13 September 2022. The absence of standard treatment and the appearance of mutated strains of SARS-CoV-2 might render the developed vaccine inactive, therefore, a quick spread of infection among patients admitted to hospitals is expected [6]. This puts healthcare workers (HCWs) at a higher risk of getting the infection due to shedding of the infectious agents, from either infected patient or carrier patients to HCWs, who further spread the infection among others in the healthcare setting, as has already been reported [7,8,9].

The World Health Organization (WHO) advised people to adopt a healthy lifestyle, in addition to following prevention and control measures, to control HAIs including SARS-CoV-2 [6]. The latter involved keeping social distance, wearing masks, and keeping hand hygiene. Hand hygiene was reported to be a critical factor for controlling the spread of infections among HCWs and patients in hospital care settings [6,10,11]. Although washing hands with soap and water are effective for keeping hand hygiene, alcohol-based hand sanitizers are commonly used for hand disinfection in hospital settings due to their broad-spectrum antimicrobial activity against bacteria, viruses, parasites, and fungi [12,13]. However, frequent administration of alcohol-based hand sanitizers has been reported to be associated with several side effects, such as redness and cracking of skin, with the possibility of infection, as well as developing microbial resistance and virus outbreaks [6,11,14,15,16,17,18,19]. Based on this, it is highly recommended to find alternative approaches to replace alcohol-based hand sanitizers. Rhamnolipids are biosurfactants produced by *Pseudomonas aeruginosa* and could be produced efficiently and economically at industrial scale [20,21,22]. We previously reported the antibacterial activity of rhamnolipids nano-micelles against selected resistant Gram-negative and Gram-positive bacteria, and a docking study showed rhamnolipids nano-micelles to be effective against SARS-CoV-2 [6,23]. In addition, an in-vitro study confirmed the superior safety of rhamnolipids nano-micelles against human dermal fibroblast cells over alcohol-based hand sanitizer [23]. Thus, we previously recommended rhamnolipids nano-micelles to replace alcohol-based hand sanitizer for hand sanitization [23].

The antiviral activity of rhamnolipids mixtures was previously reported against the Severe Acute Respiratory Syndrome (SARS-CoV-2) strain VR PV10734 clinical isolate, CoV-229E (ATCC VR-740), HCoV-OC43 (ATCC VR-1558), HSV-1 strain SC16, fluorescent HSV-1 (GFP-HSV-1), HSV-2 strain 333, Poliovirus Type 1 (PV-1) strain Chat (ATCC VR-1562) [5]. However, the antiviral activity of rhamnolipids nano-micelles against SARS-CoV-2 has not yet been studied. Herein, we investigated the antiviral activity of rhamnolipids nano-micelles against replicating SARS-CoV-2. To expand on the previous experiments addressing the safety of rhamnolipids nano-micelles in-vitro on human dermal fibroblast cells [23], we also performed skin and eye irritation tests for rhamnolipids nano-micelles on rabbits.

## 2. Results

### 2.1. Production of Rhamnolipids

The successful production of rhamnolipids was confirmed using an ESI-MS spectrometer coupled with UPLC (LC/ESI-MS) in our previous publication [6,23]. The obtained rhamnolipids were composed of a higher proportion of mono-rhamnolipids to di-rhamnolipids (Appendix A).

### 2.2. Preparation and Characterization of Rhamnolipids Nano-Micelles

Rhamnolipid nano-micelles solution was prepared as described in Section 4.2.2. The particle size and zeta potential were recorded using a Malvern Zeta-sizer instrument (Malvern Instruments Ltd., Malvern, Worcestershire, UK), and are presented as an average diameter (D, nm) ± SD and an average zeta potential (mv) ± SD, respectively. The particle size of the nano-micelles solution prepared at 20, 78, and 312 µg/mL were 191 ± 22.16, 265 ± 33.56, and 188 ± 46.98, respectively. The polydispersity index (PDI) values were 0.33, 0.30, and 0.36 for samples prepared at concentration, 20, 78, and 312 µg/mL, respectively, and this is indicative of a monodisperse sample. The concentration of rhamnolipids had a non-significant (*p* > 0.05) effect on particle size. Zeta potential values were −47.93 ± 1.81, −35.23 ± 4.32, and −41.57 ± 13.70 for samples prepared at 20, 78, and 312 µg/mL, respectively, indicating highly stable samples.

Transmission electron microscopy (TEM) images of rhamnolipid nano-micelles solution prepared at 20, 78, and 312 µg/mL are shown in Figure 1. The images showed spherical nano-micelles with no sign of aggregation, and the particle size ranged from 84.5 to 95.5 nm, from 80.8 to 105, and from 76.5 to 114 nm for nano-micelles prepared at 20, 78, and 312 µg/mL, respectively. However, samples prepared at 20 µg/mL showed some particles of lower size. The particle size identified with TEM was smaller than that recorded with the Malvern Zeta-sizer instrument.

The pH for rhamnolipid nano-micelles prepared at different concentration of rhamnolipids was determined as described in Section 4.2.3 where pH value of all solution was very close to each other and was equivalent to 6.32 ± 0.02, 6.22 ± 0.05, and 6.43 ± 0.1 for samples prepared at 20, 78, and 312 µg/mL, respectively.

### 2.3. Antiviral Activity of Rhamnolipids Nano-Micelles

The antiviral activity of rhamnolipids nano-micelles against SARS-CoV-2 are presented in Figure 2. Vero cells are the gold standard for coronavirus infection in vitro and support the full virus life cycle for SARS-CoV-2. To assess antiviral activity of rhamnolipids nano-micelles, we infected Vero cells with SARS-CoV-2 Wuhan (Public Health England), as described in Section 4.2.4. The virus was pre-treated for 1 h with blank PBS (10 mM, pH 7.4) solution, a vehicle was used to disperse rhamnolipids nano-micelles or with the indicated concentrations of nano-micelles. Representative images of infected cells are shown in Figure 2A, and quantification of the results in Figure 2B. There was no cytotoxic effect noted at the tested concentrations. Rhamnolipids nano-micelles solution at 20, 78, and 312 µg/mL all demonstrated a significant (*p* < 0.05) decrease of virus infectivity compared to virus only and the blank solution sample.

### 2.4. Irritation and Skin Sensitization Test

#### 2.4.1. Skin Irritation Test

In the skin irritation test, the skin was observed for signs of irritation, e.g., erythema/edema based on the Magnusson and Kligman scale [24]. The items of skin irritation on the scoring charts consisted of erythema, eschar, and edema. The degree of irritation on the skin was checked twice over three days, once at 24 h and once at 72 h. The obtained results, presented in Table 1, demonstrate the absence of any differences between untreated (control) and treated sites for both abraded and intact skin regarding all items checked: erythema, eschar, and edema. Additional data concerning skin irritation was provided in Appendix A.

During the test, the control sites for the abraded skin were more reddened compared to intact skin due to being scratched with the needle of the syringe, however, after 45 min, the redness disappeared, and the skin returned to its original state. As presented in Figure 3, control (untreated) and treated sites for both abraded and intact skin were lacking any abnormal signs, where eschar, edema, and erythema were not observed on the back of the rabbits. The photos presented in Figure 3 were taken at the end of the test (after 72 h) for the intact and abraded sites of untreated site (control) and sites treated with rhamnolipids nano-micelles (0.625 mg/mL), and PBS (10 mM, pH 7.4). Additional images presenting skin irritation was provided in Appendix A.

#### 2.4.2. Eye Irritation Test

The observations of eye cytotoxicity were conducted at 24 and 72 h after initiating the test for both the untreated animal and for the animal treated with tested solutions of rhamnolipids nano-micelles (0.625 mg/mL), PBS (10 mM, pH 7.4), the vehicle used to disperse nano-micelles. The assessment of eye irritation was conducted according to the histological grading system for eye irritation [25]. Different grading from 0 to 3 was carried out on the basis of the absence or presence, and severity of, symptoms, including inflammation, redness, and tearing. The scoring chart of the guideline involved checking the following: corneal opacity, reactivity of iris, chemosis, and discharge of eyes after the sample instillation into the eye (treated sites). The obtained data are presented in Table 2. The instillation of samples into the eye was not associated with any irritation or abnormality in the treated eye after 24 and 72 h of sample application compared to the untreated eye. Additional data concerning eye irritation was provided in Appendix A.

The untreated (negative control) and treated eyes were imaged after 72 h and the images are presented in Figure 4. The pupil sizes differ due to the varying light levels while taking photos. As can be seen, the images of the treated eyes, in comparison to the untreated eyes, demonstrated no differences between them. This indicates the absence of any abnormality that might occur to the cornea, iris, and conjunctiva due to treatment with tested solutions, rhamnolipids nano-micelles (0.625 mg/mL) and PBS (10 mM, pH 7.4). Additional images presenting eye irritation was provided in Appendix A.

#### 2.4.3. Skin Sensitization Test

In the skin sensitization test, the skin was observed for signs of irritation, e.g., erythema/edema based on the scoring system of the Magnusson and Kligman grading score [24]. These parameters were checked once at 24, 48, and 72 h after the intradermal injection of tested solutions, and the results are presented in Table 3 and Figure 5. The obtained results revealed that animals treated with rhamnolipids nano-micelles (0.625 mg/mL) showed no clinical signs of sensitization and were identical to the negative control, contrary to marked signs of skin sensitization identified with positive control. This included redness, edema and erythema.

#### 2.4.4. Histopathological and Immunohistochemical Evaluations

The skin, eyes, lungs, and liver of the animals treated with rhamnolipids nano-micelles (0.625 mg/mL), and PBS (10 mM, pH 7.4) were investigated for any histopathological changes in comparison to the untreated animal. The histopathological features for skin are presented in Figure 6 and Figure 7.

The skin histopathology conducted after the skin irritation test are presented in Figure 6 (additional images were also presented in Appendix A), and revealed that both the treated and untreated animals showed normal skin histology where skin layers appeared normal with the absence of any sign of erosion, ulcers, necrotic cells, or inflammatory cells. The histopathological images of the skin sensitization test are presented in Figure 7. As revealed, positive control (Figure 7A) showed that the structure of dermis and epidermis was thickened and disrupted, with some areas of necrosis with infiltration of pleomorphic inflammatory cells. This is in contrast to the animals injected intradermally with rhamnolipids nano-micelles (0.625 mg/mL) (Figure 7B), where skin epidermis and dermis appeared completely normal, and showed no histopathological alterations compared to the negative control group (Figure 7C).

The histopathological features for the eyes, lungs and liver after the irritation test are presented in Figure 8, Figure 9 and Figure 10.

For the eye histopathological examination (Figure 8), the cornea and fibrous connective tissue of the treated and untreated animals appeared normal, with no signs of inflammation, erosion, ulcers or necrobiotic changes. The ciliary body also appeared normal, with no cutting in the filament of the ciliary body nor oedema. Additional images concerned with eye histopathology were also presented in Appendix A.

In terms of the lung histopathological examination of both the treated and untreated animals (Figure 9), both showed normal bronchioles and alveolar structure. Liver histopathology images of both treated and untreated animals (Figure 10) revealed normal periportal regions, normal hepatocyte morphology, with no evidence of inflammation or necrosis.

### 2.5. Biochemical Analysis

Liver function biomarkers were analyzed in serum samples of both the treated and untreated animals, and the data are presented in Table 4. The levels of alanine aminotransferase (ALT) and aspartate aminotransferase (AST) measured for the animals treated with rhamnolipids nano-micelles (0.625 mg/mL) and PBS (10 mM, pH 7.4), were non-significantly (*p* > 0.05) different than the levels of ALT and AST measured in the untreated animals.

### 2.6. Hematological Examination

The hematological examinations for the animals treated with rhamnolipids nano-micelles (0.625 mg/mL), PBS (10 mM, pH 7.4) and the untreated animals are presented in Table 5. The mean values of Erythrogram for the treated animal were non-significantly (*p* > 0.05) different than those determined for the untreated animal.

## 3. Discussion

Nanotechnology is a cutting-edge science that is known to change the physicochemical properties of materials due to their small size (1–1000 nm) and thus potentiate materials’ activity and overcome their side effects [26,27,28,29,30,31,32,33,34,35]. Nanoparticles were reported to be applied for antimicrobial purposes against viruses [6,36,37,38,39] and bacteria [6,23,40].

Rhamnolipids mixtures are biosurfactants produced by *Pseudomonas aeruginosa* and were reported to have antibacterial and antiviral activity [41,42,43,44]. We applied nanotechnology to formulate rhamnolipids nano-micelles and demonstrated their antibacterial activity against selected resistant bacterial strains: *Staphylococcus aureus*, *Streptococcus pneumoniae*, *Salmonella* Montevideo, *Salmonella* Typhimurium, and *Acinetobacter baumannii* [6,23]. The MICs recorded for rhamnolipids nano-micelles against tested bacteria implied superior antibacterial activity when compared to MICs of rhamnolipids mixtures reported in the literature [41,45]. For example, the MIC recorded for rhamnolipids mixtures with different resistant strains of *Staphylococcus aureus* ranged from 128 to 650 µg/mL [41,45], while the MIC recorded for the resistant strain of *Staphylococcus aureus* treated with rhamnolipids nano-micelles ranged from 31 to 39 µg/mL [6,23]. Similarly, the MIC recorded for rhamnolipids mixtures with *Streptococcus pneumonia* (clinically isolated) was 128 µg/mL [41], compared to 31 µg/mL, recorded for rhamnolipids nano-micelles [6]. Furthermore, we reported the superior in-vitro biosafety of rhamnolipids nano-micelles compared to alcohol-based hand sanitizer on human dermal fibroblast cells [23]. Although rhamnolipids mixtures demonstrated an antiviral activity against several viruses, including SARS-CoV2 [5,44], according to the author’s knowledge, the antiviral activity of rhamnolipids nano-micelles has not yet been investigated.

In the current study, rhamnolipids were produced from the *P. aeruginosa* strain LeS3, and then rhamnolipid nano-micelles were prepared following our previously reported protocol [6,23]. In our previous studies [6,23], we reported that the maximum MIC value recorded against the resistant strain was 312 µg/mL, thus, this concentration should be used to ascertain the complete eradication of resistant bacterial strain. As viruses have a different nature and cell structure than bacteria, we prepared rhamnolipids nano-micelles at different concentrations of rhamnolipids, 20, 78, and 312 µg/mL, and investigated their antiviral activity against SARS-CoV-2. The different concentrations did not affect the particle size of nano-micelles produced, as revealed from the Malvern Instrument analysis and TEM images. However, the TEM images showed a lower size than that recorded by the Malvern Instrument analysis, and this could be attributed to the different techniques applied, and is consistent with what is reported in the literature [6,23,46]. All samples were homogenously distributed as the recorded PDI values were around 0.3. This is consistent with our previous publications [6,23], and with the literature [47,48,49]. All samples were stable and had a lower tendency to aggregate, as revealed from the zeta potential values, which were >−30 mv [6,23]. The pH value for all of the nano-micelles samples ranged from 6.22 to 6.43; this is consistent with the acceptable pH range (4–7) recommended for skin application [50,51] and is not likely to induce any skin irritation, as previously reported [6,23,52,53,54].

The rhamnolipids nano-micelles, at all investigated concentrations (20, 78, and 312 µg/mL), successfully inactivated the virus, and this could be explained by the significant (*p* < 0.05) reduction of viral infectivity compared to the positive control. SARS-CoV-2 infectivity was reduced by 93%, 99.4%, and 100% at rhamnolipids nano-micelles concentration of 20, 78, and 312 µg/mL, respectively. Thus, 312 µg/mL is highly recommended to be used for hand sanitation due to its complete eradication of the virus, and hence limits the spread of infection among patients and HCW.

The rhamnolipids mixtures were reported [5] to inactivate SARS-CoV-2 by 70% at 25 µg/mL. Therefore, the rhamnolipids nano-micelles demonstrated superior antiviral activity compared to the rhamnolipids mixtures. A docking study was performed in our previous publication [6] to explore the possible mechanisms of SARS-CoV-2 deactivation by rhamnolipids nano-micelles or as a singlet molecule. SARS-CoV-2 is an enveloped virus, the envelope is composed of a lipid bilayer anchored with viral spike glycoproteins that are essential for virus binding and entry into host cells [55]. The docking study suggested that rhamnolipids interaction with spike glycoproteins resulted in irreversible changes to their structures, and thus, virus deactivation. Additionally, rhamnolipids were suggested to interact with the lipid membranes (lipid envelope) of SARS-CoV-2, and this is associated with the disruption of membrane permeability, similar to that previously observed for HSV1 and HSV2 [44,56,57,58,59], and therefore, virus deactivation. Furthermore, above rhamnolipids critical micelle concentration (CMC), the lipid bilayer of SARS-CoV-2 is expected to be completely solubilized by the surfactants, and only the micellar aggregates remain in the solution [60]. Thus, the complete solubilization of the protective lipid bilayer leads to the potential disintegration of the virus into fragments, neutralizing its infectivity. Alternatively, nano-micelles are able to completely entrap the viral particle internally via hydrophobic—hydrophobic interactions [61].

Based on the previous study we performed [23], the MIC values recorded for rhamnolipids nano-micelles solution against tested multidrug resistant Gram-positive and Gram-negative bacteria ranged from 31 to 39 and 312 to 512 µg/mL, respectively [6,23]. The cytotoxic concentration of the rhamnolipid nano-micelles solution responsible for the death of 50% of human dermal fibroblast cells (CC50) was 604 µg/mL [23]. Additionally, rhamnolipids nano-micelles were able to deactivate SARS-CoV-2 at 312 µg/mL. Taken together, rhamnolipids nano-micells at a concentration ≤600 µg/mL were demonstrated to have an effective antibacterial and antiviral activity, as well as being compatible with biological system [6,23].

The acute irritation test, conducted on rabbits, was performed to further ascertain the biosafety of rhamnolipids nano-micelles. For the eye and skin irritation tests, no degree of irritation was recorded after topical application of rhamnolipids nano-micelles (0.625 mg/mL), or after its instillation into the eye of rabbits. Additionally, the skin sensitization test performed confirmed further the compatibility of rhamnolipids nano-micelles with animal skin.

The results obtained from the histopathological, biomarker, and hematological analyses from animals treated with rhamnolipids nano-micelles (0.625 mg/mL) demonstrated an absence of any signs of irritation or sensitization and, thus, assured the high safety of rhamnolipids nano-micelles solution being applied topically for hand sanitation. The obtained data is consistent with work performed by Bharali and colleagues [62], in which an acute dermal irritation study of rhamnolipids mixtures showed no dermal reactions, such as erythema or edema, compared to their negative control at 72 h after the application of rhamnolipids mixtures on the shaved skin of rabbits. Furthermore, the concentration of biosurfactants above their CMC (5–200 mg/L) [63] was also nontoxic to the skin of a rabbit. The acute dermal irritation study with the isolated biosurfactants also showed no adverse effect on the hematological parameters for the treated rabbits compared to untreated rabbits [62].

These results suggested that rhamnolipids nano-micelles (0.625 mg/mL) had no toxicity and could be applied safely for hand sanitation. However, a pilot study in a healthcare setting, to assess the efficacy and safety of rhamnolipids nano-micelles application as a hand sanitizer, should be performed in the future to ascertain its antimicrobial efficacy and safety compared to alcohol-based hand sanitizers.

## 4. Materials and Methods

### 4.1. Materials

Microbiological media (tryptone soy broth; TSB, and tryptone soy agar; TSA) were purchased from Hi-Media, Mumbai, India. Peptone and sodium chloride were purchased from Oxoid, UK. Hydrochloric acid, ethyl acetate, and sulfuric acid were purchased from Honeywell™, Charlotte, NC, USA. l-rhamnose was purchased from Sigma-Aldrich, Darmstadt, Germany. Orcinol was obtained from SDFCL, Chennai, India. Carbopol gel, phosphate-buffered saline (PBS) tablets, and absolute ethyl alcohol were purchased from Merck, Darmstadt, Germany. All chemicals and reagents were of analytical grade.

The African green monkey kidney epithelial cell line (Vero) was purchased from the American Type Culture Collection (ATCC). Vero cells were cultured in complete Dulbeccos modified eagle medium (cDMEM), containing 10%, fetal bovine serum (FBS), 1% non-essential amino acids (NEAAs), 1% penicillin, and streptomycin and 1% l-glutamine, purchased from Thermo Fisher Scientific, Waltham, MA, USA.

### 4.2. Methodology

#### 4.2.1. Production of Rhamnolipids

The production of rhamnolipids was carried out following our previously reported protocol, using the shake-flask technique [6], and then they were further purified from the production medium by acid precipitation and organic solvent extraction [64]. The *Pseudomonas aeruginosa* strain LeS3 was grown in TSB to obtain an OD_600_ of 0.8, corresponding to a density of 8 log cfu/mL. A 250 mL Erlenmeyer flask containing 100 mL of a sterilized production medium was formulated from chicken carcass soup (CCS), containing 5% chicken fat and 0.5% NaCl. The sterilized CCS was inoculated with 1% of the overnight bacterial culture. Inoculated flasks were then incubated in an orbital shaker (Vision Scientific Co., Ltd., Bucheon, Korea. VS-8480SR) at 30 °C and 150 rpm for 5 days. At the end of the incubation period, bacterial cells were removed from the culture broth by centrifugation at 10,000 rpm and 5 °C for 10 min (Sigma, 3–6PK) to obtain cell-free supernatant (CFS). The CFS was acidified to pH 2.0 using 1N HCl and stored overnight at 5 °C. Rhamnolipids were then extracted using an equal volume of ethyl acetate. A yellow–brown viscous paste of rhamnolipids was obtained and then stored in the fridge until further use.

#### 4.2.2. Preparation of Rhamnolipids Nano-Micelles

Rhamnolipid nano-micelles were prepared following our previously published protocol [6]. An aqueous solution of rhamnolipids, at a concentration of 10 mg/mL, was sonicated in phosphate-buffered saline (PBS, 10 mM, pH 7.4), using a probe sonicator (Dr. Hielscher Sonicator, Teltow, Germany) to form rhamnolipid nano-micelles.

#### 4.2.3. Characterization of Rhamnolipids Nano-Micelles

##### Particle Size and Zeta Potential

The particle size and zeta potential of the rhamnolipid nano-micelle solution were determined using a Malvern Zeta-sizer Nano ZS (Malvern Instruments, Ltd., Malvern, Worcestershire, UK) at 25 °C ± 0.1. Samples were diluted in PBS to give a count rate ranging from 50 to 300 KCPs.

##### Transmission Electron Microscopy

The rhamnolipid nano-micelles hand sanitizer solution was imaged by TEM (H-700, Hitachi, Ltd., Tokyo, Japan), at an accelerated voltage of 80 kV, using the negative staining method. The rhamnolipid nano-micelles hand sanitizer solution was diluted (1:50) with double-distilled water, and then a drop of the diluted solution was spread on a mesh copper grid coated with carbon film and kept for 5 min to dry. Then, a drop of phosphotungstic acid (2% *w*/*v*) was added to the grid for 50 s, and the excess liquid was removed using filter paper.

##### Determination of pH

The pH of the rhamnolipid nano-micelles were determined at room temperature. The pH was determined using an Ohaus Economical pH bench meter (starter 3100, New Haven, CT, USA) that was previously calibrated with three standard buffer solutions (pH of 4, 7, and 10).

#### 4.2.4. SARS-CoV-2 Infection Experiments (HCoV-19/England/2/202 Strain)

The Vero cells (ATCC^®^ CCL-81) were washed with PBS, dislodged with 0.25% Trypsin- EDTA (Sigma Life Sciences) and seeded into 96-well imaging plates (Greiner, Gloucestershire, UK) at a density of 8 × 10^3^/well in culture media (DMEM containing 10% FBS, 1% Penicillin and Streptomycin, 1% l-Glutamine and 1% non-essential amino acids). The next day, cells were infected with SARS-CoV-2 strain hCOV-19/England/2/2020. Virus stock 10^6^ IU/mL (kind gift from Christine Bruce, Public Health England) was diluted 1/150 in culture media allowing 25 μL per well. The virus was then diluted further, with 25 μL per well media containing treatments of interest prepared at 2×concentration to give a final at 3333 IU/mL for infection. The virus-nano-micelles mixture was incubated for 1 h at room temperature before being added to Vero cells for 24- or 48-h infections. The cells were then blocked in PBS containing 10% FBS and stained with rabbit anti-SARS-CoV-2 spike protein, subunit 1 (The Native Antigen Company, Oxford, UK), followed by Alexa Fluor 555-conjugated goat anti-rabbit IgG secondary antibody (Invitrogen, Thermo Fisher Scientific, Waltham, USA). The cell nuclei were stained with Hoechst 33342 (Thermo Fisher Scientific). After being washed with PBS, the cells were imaged and analyzed using a Thermo Scientific CelIInsight CX5 High-Content Screening (HCS) platform. Viable cells were counted, and infected cells were scored by spike perinuclear fluorescence above a set threshold determined by positive (untreated) and negative (uninfected) controls. A minimum of nine fields of view and 5,000 nuclei per well in triplicate wells per treatment were scored in each experiment. All experiments were repeated 2–4 times.

#### 4.2.5. Irritation and Skin Sensitization Study

##### Animal

New Zealand White rabbits (CLAVCAP-VACSER, Cairo, Egypt), weighing 2.5–3 kg, were used to investigate the irritation of rhamnolipids nano-micelles solution on the eyes and skin. Rabbits were maintained under managed conditions: 12 light-dark cycle, 25 ± 2 °C and 50 ± 20% relative humidity. They had free access to food, a standard commercial pellet diet (containing at the very least; 5% fiber, 20% protein, 3.5% fat, 6.5% vitamins and ash mixture) and were offered water and libitum. The rabbits were kept in hygienic conditions throughout the experimental period, and they were left for one week to acclimatize to the lab conditions, before the onset of the experiment. All the procedures in this research were approved by the Faculty of Veterinary Medicine, Cairo University (Vet CU20092022530). All efforts were made to minimize the animals suffering.

##### Eye Irritation Test

The Draize modified test was carried out to determine ocular irritation in the eyes of ten white Zealand rabbits after the application of rhamnolipids nano-micelles and PBS (10 mM, pH 7.4), where five rabbits were used to test each solution [65]. The rhamnolipids nano micelles solutions and PBS (50 µL) were instilled into the lower cul-de-sac of the rabbits’ eye, to be exposed to cornea. The rhamnolipids nano-micelles were administrated five times in the left eye, with each dose separated by 5 min time intervals. The right eye served as a negative control (untreated eye). Eyelids were gently kept together for around 10 s to prevent the loss of the instilled preparation. After dose instilment, the rabbit’s eyes were observed for any possible ocular reactions, including redness of the eyes, conjunctival chemosis, discharge, and corneal and iris lesions. These observations were conducted at regular time intervals of 24 and 72 h. The assessment was conducted according to the histological grading system for eye irritation evaluation [25]. Different grading, from 0 to 3, was carried out on the basis of the absence or presence, and severity, of symptoms, including inflammation, redness, and tearing [66]. Photos were taken for all groups after 72 h of treatment.

##### Skin Irritation Test

Six rabbits were anesthetized to remove the hair on an area of their backs. Six squares, with a dimension of 2.5 × 2.5 cm^2^, were marked in the glabrous area on each rabbit. The three squares on the left were scratched with a syringe needle (abraded skin); the three squares on the right side were not scratched (intact skin). The upper two squares on each side act as negative control for abraded and intact skin. On the middle squares of each side, rhamnolipids nano-micelles solution (0.5 mL) was applied. PBS (10 mM, pH 7.4) was the vehicle used to disperse the nano-micelles, thus, on the lower two squares, PBS (10 mM, pH 7.4, 0.5 mL) was applied. All of the squares were covered with sterile gauze of the same size as the square, and nonirritant tape was used to fix the gauze in place. After 24 and 72 h, the skin was observed for signs of irritation, e.g., erythema/edema, based on the Magnusson and Kligman scale [24]. Photos were taken for all groups after 72 h of treatment.

##### Skin Sensitization Test

The test was carried out on healthy albino guinea-pigs, in accordance with the OECD guideline [67]. Twenty healthy albino guinea-pigs were considered for a sensitization test, and were kept in individual cages for 4 days under 20 °C ± 2 °C, 50 ± 10% relative humidity, natural illumination, with conventional diet and water, with sufficient quantity of ascorbic acid to adjust themselves to the environment. The acclimatized albino guinea-pigs were divided into three groups: the positive control (5 animals) animals received intradermal injections (0.1 mL) of 1:1 mixture (*v*/*v*) Freund’s Complete Adjuvant (FCA)/physiological saline as skin sensitizing agent; the tested group (10 animals) animals received intradermal injections (0.1 mL) of Rhamnolipids nano-micelles solution (0.625 mg/mL); and the negative control (5 animals) received intradermal injections of PBS (0.1 mL).

Challenge reactions were assessed at 24, 48 and 72 h after intradermal injection of the samples. The intensity of all skin reactions was graded following the sensitization Magnusson and Kligman grading score [24].

##### Histopathological and Immunohistochemical Evaluations

After 72 h of the treatment protocol, the animals were sacrificed by ether anesthesia and the treated dorsal area was excised and rinsed with ice cold phosphate buffer saline. Next, the skin, eyes, liver, and lungs were fixed in 10% buffered formalin. In a routine manner, formalin fixed skin, eye, liver and lung specimens were dehydrated in different grades of alcohol, followed by clearance in xylol and embedding in paraffin. Serially, sections of 4–5 μm thickness were obtained from the prepared paraffin blocks, followed by their staining with Hematoxylin and Eosin (H & E) [68]. Histopathological findings in the skin were graded semi-quantitatively, according to [69], with some modifications (0 = no abnormality, 1 = slight, 2 = mild, 3 = moderate) for each of the seven findings: hypertrophy, hyperkeratosis, parakeratosis, erosion, inflammatory cells infiltration, extracellular edema and ulcer. Histopathology of skin samples was carried out at the Faculty of Veterinary Medicine, Cairo University, Egypt.

##### Biochemical Analysis

Liver function biomarkers were analyzed in serum samples of rabbits of all groups, including alanine aminotransferase (ALT) and aspartate aminotransferase (AST) using kits of EIAab^®^ (Wuhan, China).

##### Hematological Examination

Complete blood pictures (CBC) were obtained using ABC animal blood count apparatus (vet 907 AB 6012).

#### 4.2.6. Statistical Analysis

Statistical analysis was performed using two-way ANOVA. Analyses were carried out using GraphPad Prism 9.0 software at a confidence level of 95%.

## 5. Conclusions

Hospital- and community-acquired infections are escalating and pose a serious public health problem worldwide. Hands are considered to be an important route for transmitting microbes and infections between individuals. Thus, keeping good hand hygiene is a key factor to control or prevent the spread of infection. Due to reports of skin cytotoxicity caused by the frequent use of alcohol-based hand sanitizer during the COVID-19 pandemic, it was essential to develop alternative and novel approaches for hand sanitation. In our previous publications, we demonstrated the antibacterial activity of rhamnolipids nano-micelles against selected multidrug resistant Gram-positive and Gram-negative bacteria and reported the maximum MIC value to be 312 µg/mL. In the current work, rhamnolipids nano-micelles solution, at 20, 78, and 312 µg/mL, all demonstrated a significant (*p* < 0.05) decrease of SARS-CoV-2 virus infectivity compared to virus only and the blank sample. However, 100 % virus eradication was only recorded with 312 µg/mL. The acute irritation test and skin sensitization test revealed that rhamnolipids nano-micelles were biocompatible with the skin. Furthermore, histopathological studies on skin after the irritation test revealed normal skin histopathology, where skin layers appeared normal with the absence of any signs of erosion, ulcers, necrotic cells, or inflammatory cells. The compatibility of rhamnolipids nano-micelles with the skin was further confirmed by the skin sensitization test, where layers of epidermis and dermis appeared completely normal, with no histopathological alterations in comparison to negative control group. Thus, rhamnolipids nano-micelles are recommended to be used as a safe and effective hand sanitizer. However, a future study in hospitals is still to be performed to ascertain their broad-spectrum antimicrobial efficacy and compatibility with skin.

## Figures and Tables

**Figure 1 antibiotics-11-01556-f001:**
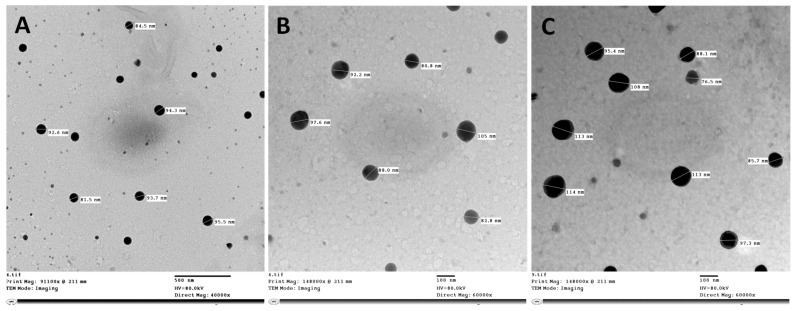
TEM images of rhamnolipid nano-micelles prepared at rhamnolipids concentration (**A**) 20, (**B**) 78, and (**C**) 312 µg/mL, respectively.

**Figure 2 antibiotics-11-01556-f002:**
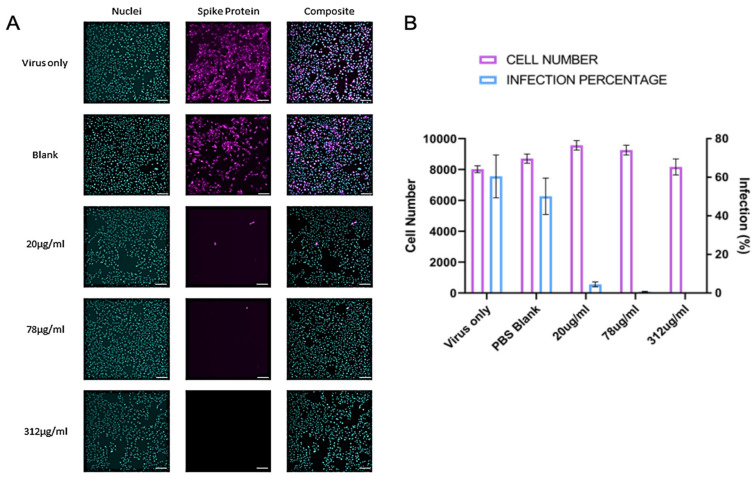
Antiviral activity of rhamnolipids nano-micelles solution prepared at 20, 78, and 312 µg/mL against SARS-CoV-2. (**A**) Representative immunofluorescence images for infected Vero cells where nuclei were detected by Hoechst 33342 (Cyan) and SARS-CoV-2 spike protein by rabbit anti-spike antibody detected with anti-rabbit-Alexa-555 secondary antibody (magenta). Scale bars represent 100 µm. (**B**) Automated quantification of cell number per field of view and % infected cells 48 h after infection. Results are the average of three independent experiments, with three replicates in each. Error Bars represent standard deviation.

**Figure 3 antibiotics-11-01556-f003:**
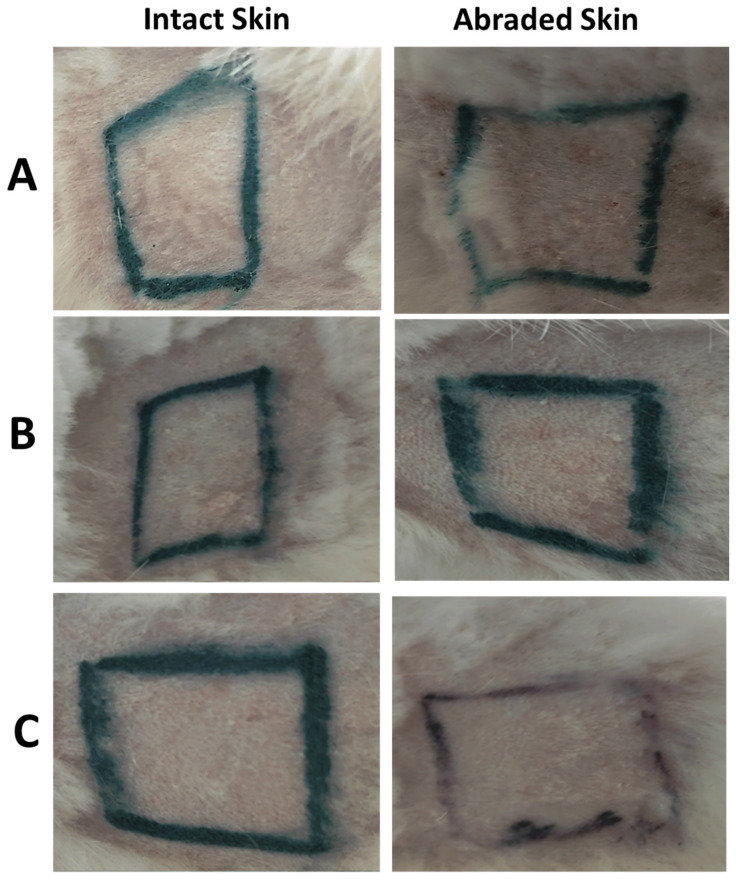
Representative photos of the skin irritation test for one representative rabbit for (**A**) Group I; untreated group, act as a control, (**B**) Group II; rhamnolipids nano-micelles solution (0.625 mg/mL) dispersed in PBS (10 mM, pH 7.4) treated group, and (**C**) Group III; PBS (10 mM, pH 7.4) treated group. Black lines were drawn with a non-irritating pen. No differences in score were observed after 72 h between the test and control site for both abraded and intact skin.

**Figure 4 antibiotics-11-01556-f004:**
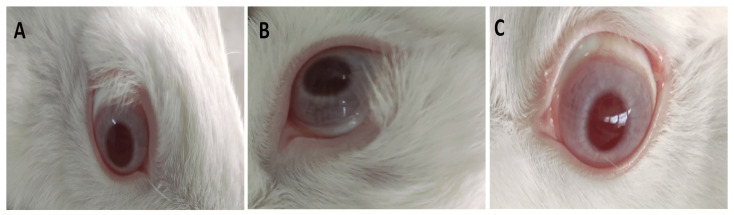
Representative photos of the eye irritation test for one representative rabbit for (**A**) Group I; untreated group, act as a control, (**B**) Group II; PBS (10 mM, pH 7.4) treated group, and (**C**) Group III; rhamnolipids nano-micelles solution (0.625 mg/mL) dispersed in PBS (10 mM, pH 7.4) treated group. The cornea, iris, and conjunctiva were observed after 72 h. In this study, no differences of cornea, iris and conjunctiva were observed among different groups.

**Figure 5 antibiotics-11-01556-f005:**
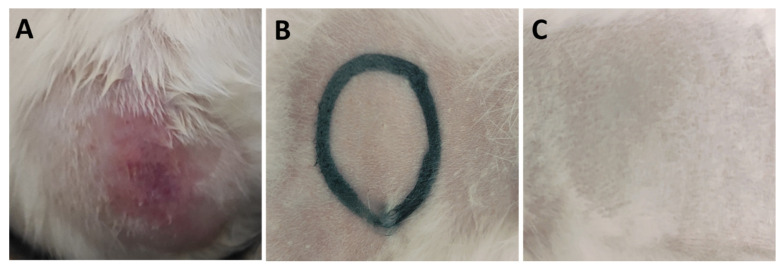
Representative photos of skin sensitization test for one representative guinea pigs for (**A**) Positive control; animals injected intradermally with skin sensitizing agent, a mixture of Freund’s Complete Adjuvant (FCA), and physiological buffered saline (PBS) (1:1 mixture, 0.1 mL) (**B**) Rhamnolipids nano-micelles solution (0.625 mg/mL) dispersed in PBS (10 mM, pH 7.4), and (**C**) Negative control; animals injected intradermally with physiological buffered saline (PBS, 0.1 mL). Black lines were drawn with a non-sensitizing or irritating pen. Rhamnolipids nano-micelles showed no clinical signs of skin sensitization (no erythema nor edema) that was identical to negative control contrary to positive control where marked signs of sensitization (erythema, edema, and redness) were identified.

**Figure 6 antibiotics-11-01556-f006:**
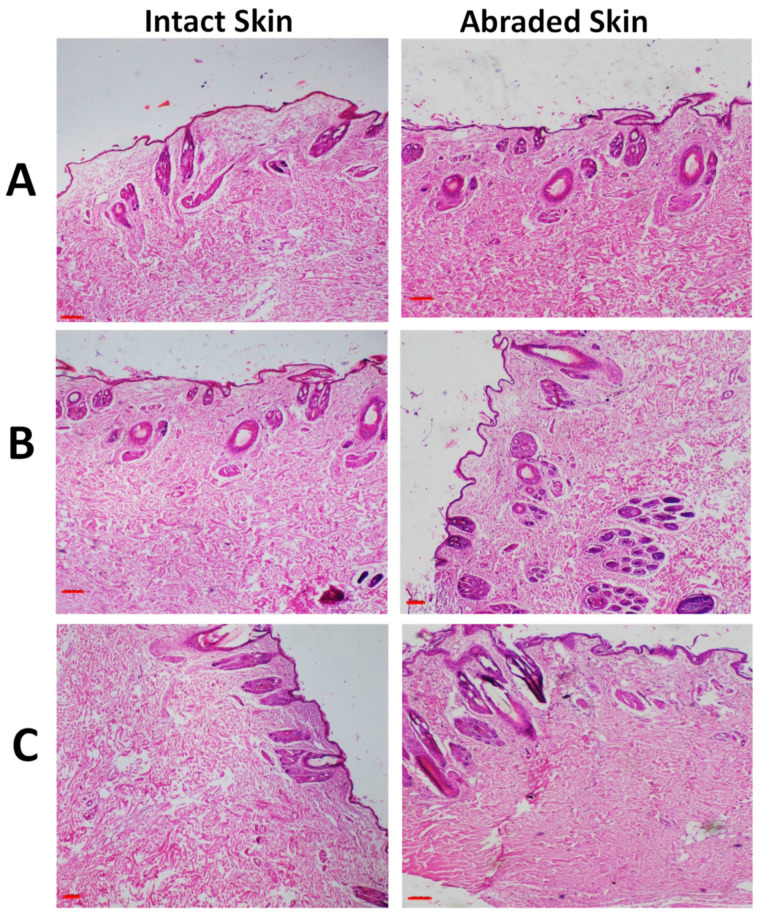
Representative images of H and E stained photomicrographs of skin for one representative rabbit after skin irritation test for (**A**) Group I; untreated group, act as a control, (**B**) Group II; Rhamnolipids nano-micelles solution group, and (**C**) Group III; PBS (10 mM, pH 7.4) treated group. All groups treated with samples showed normal histological tissue identical to untreated group after 72 h of treatment with samples. All skin layers appear normal and similar to control with the absence of any sign of erosion, ulcers, necrotic cells, or inflammatory cells. Scale bar, 100 µm.

**Figure 7 antibiotics-11-01556-f007:**
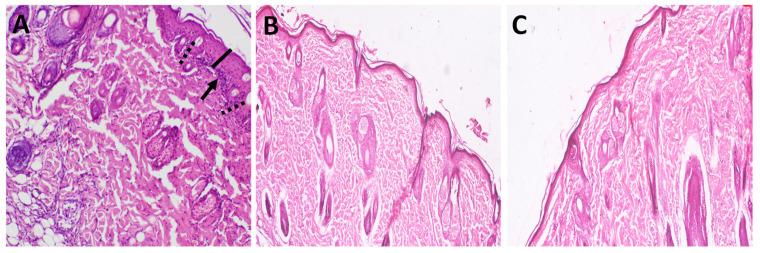
Representative images of H and E stained photomicrographs of skin for one representative guinea pigs after skin sensitization test for (**A**) Positive control; animals injected intradermally with skin sensitizing agent, a mixture of Freund’s Complete Adjuvant (FCA), and physiological buffered saline (PBS) (1:1 mixture, 0.1 mL) (**B**) Rhamnolipids nano-micelles solution (0.625 mg/mL) dispersed in PBS (10 mM, pH 7.4), and (**C**) Negative control; animals injected intradermally with physiological buffered saline (PBS, 0.1 mL). Positive control revealed thickening of epiderma (black line), inflammatory cells infiltration (black head arrow) and spongiosis (dashed line). (**B**,**C**) group revealed normal skin layers.

**Figure 8 antibiotics-11-01556-f008:**
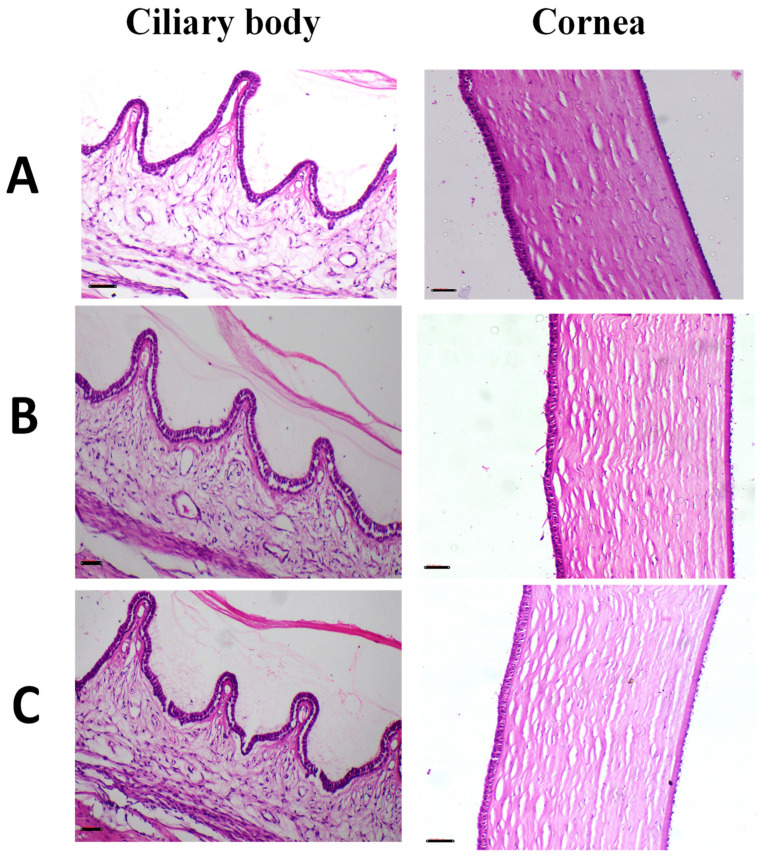
Representative images of H and E stained photomicrographs of eye for one representative rabbit after irritation test for (**A**) Group I; untreated group, act as a control, (**B**) Group II; rhamnolipids nano-micelles solution dispersed in PBS (10 mM, pH 7.4) treated group, and (**C**) Group III; PBS (10 mM, pH 7.4) treated group. All groups treated with samples showed normal histological tissue identical to untreated group after 72 h of treatment with samples. Cornea, fibrous connective tissue appeared normal with no signs of inflammation, erosion, ulcers or necrobiotic changes. Ciliary body appeared also normal with no cutting in the filament of ciliary body with no oedema. Scale bar, 50 µm.

**Figure 9 antibiotics-11-01556-f009:**
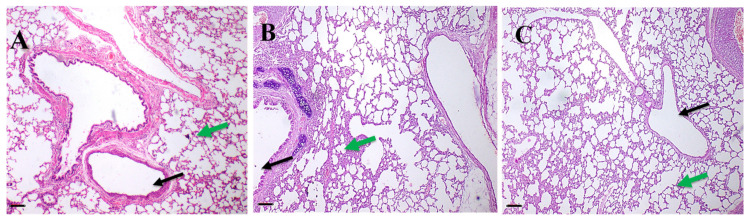
Representative photos of H and E stained photomicrographs of lung for one representative rabbit after irritation test for (**A**) Group I; untreated group, act as a control, (**B**) Group II; rhamnolipids nano-micelles solution dispersed in PBS (10 mM, pH 7.4) treated group, and (**C**) Group III; PBS (10 mM, pH 7.4) treated group. All groups treated with samples showed normal histological tissue identical to untreated group after 72 h of treatment with samples. Black arrows pointed to bronchioles while green arrows pointed to alveolar cells. Scale bar, 50 µm.

**Figure 10 antibiotics-11-01556-f010:**
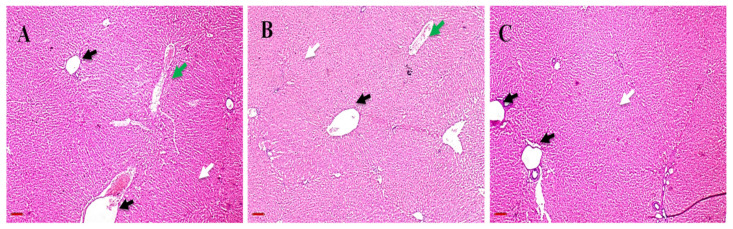
Representative images of H and E stained photomicrographs of liver for one representative rabbit after irritation test for (**A**) Group I; untreated group, act as a control, (**B**) Group II; rhamnolipids nano-micelles solution dispersed in PBS (10 mM, pH 7.4) treated group, and (**C**) Group III; PBS (10 mM, pH 7.4) treated group. All groups treated with samples showed normal histological tissue identical to untreated group after 72 h of treatment with samples. The black arrow points to hepatic central vein, the green arrow points to portal triads, and the white arrows point to hepatocytes. Scale bar, 100 µm.

**Table 1 antibiotics-11-01556-t001:** Skin irritation evaluation after treatment with rhamnolipids nano-micelles solution and PBS (10 mM, pH 7.4), vehicle used to disperse rhamnolipids nano-micelles *.

		Control Site								Treated Site							
		Erythema and Eschar				Edema				Erythema and Eschar				Edema			
		Intact		Abraded		Intact		Abraded		Intact		Abraded		Intact		Abraded	
Tested sample/tested time (h)	Animal Number	24	72	24	72	24	72	24	72	24	72	24	72	24	72	24	72
Rhamnolipids nano-micelles (0.625 µg/mL)	1	0	0	0	0	0	0	0	0	0	0	0	0	0	0	0	0
	2	0	0	0	0	0	0	0	0	0	0	0	0	0	0	0	0
	3	0	0	0	0	0	0	0	0	0	0	0	0	0	0	0	0
PBS (10 mM, pH 7.4)	4	0	0	0	0	0	0	0	0	0	0	0	0	0	0	0	0
	5	0	0	0	0	0	0	0	0	0	0	0	0	0	0	0	0
	6	0	0	0	0	0	0	0	0	0	0	0	0	0	0	0	0

* All skin irritation ratings were 0 after 24 and 72 h, both intact and abraded skin showed no symptoms (erythema, eschar, and oedema) indicating incidence of irritation.

**Table 2 antibiotics-11-01556-t002:** Evaluation of eye irritations following treatment with rhamnolipids nano-micelles solution and PBS, vehicle used to disperse rhamnolipids nano-micelles *.

Tested Solution	Tissues Examined in the Eye	Number of Rabbits					
		1		2		3	
		RT. Untreated	LT. Treated	RT. Untreated	LT. Treated	RT. Untreated	LT. Treated
Rhamnolipids nano-micelles (0.625 mg/mL)	Cornea	0	0	0	0	0	0
	Iris	0	0	0	0	0	0
	Conjunctiva	0	0	0	0	0	0
PBS (10 mM, pH 7.4)	Cornea	0	0	0	0	0	0
	Iris	0	0	0	0	0	0
	Conjunctiva	0	0	0	0	0	0

* All eye irritation scores were 0. The observations were concerned with corneal opacity, reactivity of iris, conjunctival edema, and ocular discharge. No symptoms indicated irritations were noted.

**Table 3 antibiotics-11-01556-t003:** * Skin sensitization scores recorded for guinea pigs injected intradermally with rhamnolipids nano-micelles solution versus positive and negative control **.

Hours/ Number of Animals	Positive Control Erythema/Edema			Rhamnolipids Nano-Micelles (0.625 mg/mL) Erythema/Edema			Negative Control Erythema/Edema		
	24	48	72	24	48	72	24	48	72
**1**	0/0	2/1	2/2	0/0	0/0	0/0	0/0	0/0	0/0
**2**	0/0	2/1	2/2	0/0	0/0	0/0	0/0	0/0	0/0
**3**	0/0	2/1	2/3	0/0	0/0	0/0	0/0	0/0	0/0
**4**	0/0	3/1	2/3	0/0	0/0	0/0	0/0	0/0	0/0
**5**	0/0	3/1	3/3	0/0	0/0	0/0	0/0	0/0	0/0
**6**				0/0	0/0				
**7**				0/0	0/0				
**8**				0/0	0/0				
**9**				0/0	0/0				
**10**				0/0	0/0				

* All skin sensitization ratings were 0 after 24, 48 and 72 h for all animals treated with rhamnolipids nano-micelles solution and they were identical to negative control. This indicated the absence of any signs of sensitization and the safety of rhamnolipids nano-micelles. ** Positive control; animals injected intradermally with skin sensitizing agent, mixture of Freunds’s Complete Adjuvant (FCA), and physiological buffered saline (PBS) (1:1 mixture, 0.1 mL); Negative control; animals injected intradermally with physiological buffered saline (PBS, 0.1 mL).

**Table 4 antibiotics-11-01556-t004:** * Effect of tested solutions, rhamnolipids nano-micelles and PBS (10 mM, pH 7.4) on serum biochemical markers. Data are presented as an average ± SE, results are average of three replicates.

Animal Group	AST (U/L)	ALT (U/L)
Untreated animal (Negative Control)	30.00 ± 1.53	27.33 ± 1.20
PBS	32.70 ± 3.20	28.20 ± 2.80
Rhamnolipids nano-micelles (0.625 mg/mL)	35.00 ± 4.35	28.67 ± 3.71

* Data were analyzed by two-way ANOVA, there was non-significant differences between untreated (control) and treated animals at *p* < 0.05. (U/L) = unit/liter.

**Table 5 antibiotics-11-01556-t005:** * Effect of tested solutions, rhamnolipids nano-micelles and PBS (10 mM, pH 7.4) on the animal Erythrogram. Data are presented as an average ± SE, results are average of three replicates.

Animal Groups	PCV (%)	Hb (g/dL)	RBCs × 10^6^/µL	MCV (fL)	MCHC (%)
Untreated animal (Control)	38.53 ± 0.56	12.00 ± 0.64	8.84 ± 0.11	48.87±0.22	37.26 ± 0.46
PBS	37.90 ± 0.61	11.51 ± 0.32	7.80 ± 0.13	48.20 ± 0.19	37.3 ± 0.32
Rhamnolipids nano-micelles (0.625 mg/mL)	39.41± 0.26	11.34 ± 0.20	7.61±0.140	47.72 ± 0.26	37.28 ± 0.34

* Data were analyzed by two-way ANOVA, there was non-significant differences between untreated (control) and treated animals at *p* < 0.05. PCV, Packed cell volume, Hb = Hemoglobin (g/dL = gram/deciliter), RBCs = Red blood cells count, MCV = Main corpuscular volume (fL, femto-liter), MCHC = Main corpuscular hemoglobin concentrations.

## Data Availability

All authors are happy to share all data (including Appendix A).

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
