# Peer review of "Rhamnolipid Nano-Micelles Inhibit SARS-CoV-2 Infection and Have No Dermal or Eye Toxic Effects in Rabbits"

_antibiotics, 2022, doi:10.3390/antibiotics11111556_

Round 1

Reviewer 1 Report

The manuscript entitled “Rhamnolipid Nano-micelles inhibit SARS-CoV-2 infection and have no dermal or eye toxic effects in rabbits” by Alaa M. Ali, Harriet J. Hill, Gehad E. Elkhouly, Marwa Reda Bakkar, Nermeen R. Raya, Zania Stamataki, Yasmin Abo-zeid investigated the antiviral properties of rhamnolipid nano-micelles against SARS-CoV-2. The authors further conducted irritation tests on the eyes and skin of the animal model treated with rhamnolipid nano-micelles to examine the biosafety of the compound. Additional histopathological, biochemical, and hematological analyses were also performed. This work is a further investigation of rhamnolipid nano-micelles, which have been reported to possess antibacterial properties by the same authors. Based on their findings Alaa et al. suggest that rhamnolipid nano-micelles can substitute alcohol-based hand sanitizers due to their lower skin toxicity. Such research aimed to demonstrate the antimicrobial and antiviral potential of a non-toxic compound, which is a crucial contribution to fighting against the alarming threat of hospital-acquired infections. In my view, this paper can be published with minor improvements.

The manuscript is well-structured and cohesive. The abstract section delivers a clear overview of the purpose and progress of the conducted research. The introduction section identifies the current situation concerning hospital-acquired infections and the importance of prevention, especially regarding hand hygiene. Despite being considered an effective antimicrobial method, alcohol-based hand sanitizers cause various skin complications due to multiple applications. According to the authors, rhamnolipid nano-micelles are less harmful than alcohol-based hand sanitizers yet equally potent against bacteria and viruses.

The design and methodology are reasoned throughout. Figures and tables are easily interpretable and understandable. The obtained results are clearly presented and thoroughly explained. Initially, rhamnolipids were produced by Pseudomonas aeruginosa and then formulated into nano-micelles. Particle size and zeta potential were measured. Images of rhamnolipid nano-micelles were captured using transmission electron microscopy. Antiviral properties of the compound with three different concentrations were examined on Vero cells. The infection rate was demonstrated using nuclei dye Hoechst 3342, rabbit anti-spike antibody with anti-rabbit-Alexa-555 secondary antibody dye. All the examined concentrations [20, 78, 312 µg/mL] significantly inhibited SARS-CoV-2 infection. The authors provided a comprehensive explanation of the antiviral mechanism of rhamnolipid nano-micelles. Furthermore, no differences were observed between the control and rhamnolipid nano-micelle applied rabbit skin and eyes. Results from hematological, biochemical, and histopathological evaluations were not significantly different than untreated samples. 

Please improve the following remarks:

1. The reason for choosing 20, 78, and 312 µg/mL is missing in the discussion.

2. Comment the reason particle sizes identified with TEM are smaller than the measurements conducted with Zeta-sizer.

3. Experiments examining the antiviral activities of rhamnolipid nano-micelles must be conducted with at least three biological replicates. It is unclear whether the repetitions were biological or technical in the methodology section. Figure 2 description gives the impression that the experiments were done in two biological replicates with three technical replicates each.

4. Magnification is 60000x on each image and scale bars are the same, but particle sizes do not match. For example, a particle with a size of 84.5 nm in image A is significantly smaller than a particle with a size of 80.8 nm in image B and 76.5 nm in image C.

5.  Subsection 2.3 was omitted.

6.  Additional experiment examining the repeated application of rhamnolipid nano-micelles on the skin and eyes of the model animal is needed.

7. The discussion is miswritten as section number 2.

8. Typos are made in the description of tables 3 and 4.

9. Typos are made in lines 31 and 34 in the discussion section.

10. Double space and no space between the number and sign in tables 3 and 4.

11.  Reference number 41 has a question mark sign in the reference section.

Author Response

Dear Prof. Dr. Nicholas Dixon,

Editor-in-Chief
Antibiotics

April 20th 2022

Re: Submission of revised paper Manuscript ID: antibiotics-1666343 titled, “Rhamnolipids nano-micelles versus alcohol based-hand sanitizer: A comparative study for antibacterial activity against hospital acquired infections and toxicity concerns”

Dear Professor, Nicholas Dixon,

We would like to thank you for your email dated March 28th 2022 enclosing the reviewers’ comments. We have carefully reviewed the comments and have revised the manuscript accordingly. Our responses to reviewer 1, reviewer 2, and reviewer 3 are given in a point-by-point manner below. Changes in the manuscript are marked in yellow color.

Reviewer comments

Reviewer 1

Open Review

English language and style

( ) Extensive editing of English language and style required
( ) Moderate English changes required
(x) English language and style are fine/minor spell check required
( ) I don't feel qualified to judge about the English language and style

Yes

Can be improved

Must be improved

Not applicable

Does the introduction provide sufficient background and include all relevant references?

( )

(x)

( )

( )

Are all the cited references relevant to the research?

( )

(x)

( )

( )

Is the research design appropriate?

( )

(x)

( )

( )

Are the methods adequately described?

( )

(x)

( )

( )

Are the results clearly presented?

( )

(x)

( )

( )

Are the conclusions supported by the results?

(x)

( )

( )

( )

Comments and Suggestions for Authors

The manuscript entitled “Rhamnolipid Nano-micelles inhibit SARS-CoV-2 infection and have no dermal or eye toxic effects in rabbits” by Alaa M. Ali, Harriet J. Hill, Gehad E. Elkhouly, Marwa Reda Bakkar, Nermeen R. Raya, Zania Stamataki, Yasmin Abo-zeid investigated the antiviral properties of rhamnolipid nano-micelles against SARS-CoV-2. The authors further conducted irritation tests on the eyes and skin of the animal model treated with rhamnolipid nano-micelles to examine the biosafety of the compound. Additional histopathological, biochemical, and hematological analyses were also performed. This work is a further investigation of rhamnolipid nano-micelles, which have been reported to possess antibacterial properties by the same authors. Based on their findings Alaa et al. suggest that rhamnolipid nano-micelles can substitute alcohol-based hand sanitizers due to their lower skin toxicity. Such research aimed to demonstrate the antimicrobial and antiviral potential of a non-toxic compound, which is a crucial contribution to fighting against the alarming threat of hospital-acquired infections. In my view, this paper can be published with minor improvements.

The manuscript is well-structured and cohesive. The abstract section delivers a clear overview of the purpose and progress of the conducted research. The introduction section identifies the current situation concerning hospital-acquired infections and the importance of prevention, especially regarding hand hygiene. Despite being considered an effective antimicrobial method, alcohol-based hand sanitizers cause various skin complications due to multiple applications. According to the authors, rhamnolipid nano-micelles are less harmful than alcohol-based hand sanitizers yet equally potent against bacteria and viruses.

The design and methodology are reasoned throughout. Figures and tables are easily interpretable and understandable. The obtained results are clearly presented and thoroughly explained. Initially, rhamnolipids were produced by Pseudomonas aeruginosa and then formulated into nano-micelles. Particle size and zeta potential were measured. Images of rhamnolipid nano-micelles were captured using transmission electron microscopy. Antiviral properties of the compound with three different concentrations were examined on Vero cells. The infection rate was demonstrated using nuclei dye Hoechst 3342, rabbit anti-spike antibody with anti-rabbit-Alexa-555 secondary antibody dye. All the examined concentrations [20, 78, 312 µg/mL] significantly inhibited SARS-CoV-2 infection. The authors provided a comprehensive explanation of the antiviral mechanism of rhamnolipid nano-micelles. Furthermore, no differences were observed between the control and rhamnolipid nano-micelle applied rabbit skin and eyes. Results from hematological, biochemical, and histopathological evaluations were not significantly different than untreated samples. 

Please improve the following remarks:

Reviewer1, Comment 1:

The reason for choosing 20, 78, and 312 µg/mL is missing in the discussion.

Response to reviewer 1, Comment 1

As we are aiming to use rhamnolipids nano-micelles as a replacement for alcohol-based hand sanitizers in the COVID-19 pandemic, we are interested in using rhamnolipids nano-micelles solution with a concentration that is sufficient to eradicate both bacteria and SARS-CoV2. In our previous publications, we figured out the concentration that is effective to inhibit the growth of selected resistant bacteria that are commonly known to cause hospital acquired infections in Egyptian hospitals and we have reported MIC value to be 312 µg/mL. Thus, we have to use this concentration to eradicate these bacteria, thus, we investigated 312 µg/mL to assure that the rhamnolipids nano-micelles is sufficient to effectively eradicate SARS-CoV2 and resistant bacteria and at the same time compatible to skin. However, we have investigated lower concentrations of rhamnolipids nano-micelles to figure out if nano-micelles can eradicate the virus at lower concentration. SARS-CoV-2 infectivity was reduced by 93%, 99.2% when they exposed to rhamnolipids nano-micelles concentration equivalent to 20, and 78 µg/mL. Thus, there is still a possibility for the virus to spread among patients and healthcare workers. Therefore, it is safer to use 312 µg/mL to assure both antiviral, and antibacterial activity.  Kindly check Discussion section, line 26 to 31.

Reviewer1, Comment 2:

  1. Comment the reason particle sizes identified with TEM are smaller than the measurements conducted with Zeta-sizer.

Response to reviewer 1, Comment 2

We have explained this in discussion section, line 35 where we have clarified that this might be attributed to different techniques applied and we have referenced some articles that are consistent with our hypothesis

Reviewer1, Comment 3:

  1. Experiments examining the antiviral activities of rhamnolipid nano-micelles must be conducted with at least three biological replicates. It is unclear whether the repetitions were biological or technical in the methodology section. Figure 2 description gives the impression that the experiments were done in two biological replicates with three technical replicates each.

Response to reviewer 1, Comment 3

Thanks for the reviewer comment, we have run 3 additional experiment to investigate the antiviral activity for rhamnolipids nano-micelles against SARS-CoV2, and we have updated Figure 2 with the new data obtained, kindly check Figure 2, Page 5

Reviewer1, Comment 4:

  1. Magnification is 60000x on each image and scale bars are the same, but particle sizes do not match. For example, a particle with a size of 84.5 nm in image A is significantly smaller than a particle with a size of 80.8 nm in image B and 76.5 nm in image C.

Response to reviewer 1, Comment 4

Thanks for reviewer comments, we have checked this back with TEM Lab and Eng. Ebtehal corrected the scale bar and apologized for this mistake - we have updated Figure 1 in the manuscript with the corrected magnification power sent from her, kindly check page 4

Reviewer1, Comment 5:

  1. Subsection 2.3 was omitted.

Response to reviewer 1, Comment 5

Do apologize for this mistake, it was a mistake in ordering the subsections and it has been corrected

Reviewer1, Comment 6

  1. Additional experiment examining the repeated application of rhamnolipid nano-micelles on the skin and eyes of the model animal is needed.

Response to reviewer 1, Comment 6

Thanks for the reviewer comments, we have performed additional tests and the results are the same, we have added extra obtained data in the supplementary file for the reviewer convenience.

Reviewer1, Comment 7

  1. The discussion is miswritten as section number 2.

Response to reviewer 1, Comment 7

We have corrected the ordering of the discussion to be section number 3.

Reviewer1, Comment 8

  1. Typos are made in the description of tables 3 and 4.

Response to reviewer 1, Comment 8

Thanks for the reviewer’s comments, all Typos were corrected

Reviewer1, Comment 9

  1. Typos are made in lines 31 and 34 in the discussion section.

Response to reviewer 1, Comment 9

Thanks for the reviewer’s comments, all Typos were corrected

Reviewer1, Comment 10

  1. Double space and no space between the number and sign in tables 3 and 4.

Response to reviewer 1, Comment 10

All were presented as double space line and the table numbers were upgraded to be Table 4 and 5

 Reviewer1, Comment 11

  1. Reference number 41 has a question mark sign in the reference section.

Response to reviewer 1, Comment 10

The questions mark has been deleted

Reviewer 2 Report

The current manuscript is focused on rhamnolipids nano-micelles solution demonstrated a significant decrease of SARS-CoV2 virus infectivity compared to virus only and the blank sample. Moreover, acute irritation test revealed that rhamnolipids nano-micelles are biocompatible with skin after topical application and proved to have no toxic effect on eye tissues after being instilled into the eye.

Comments

More literature should be added about rhamnolipids more effective over alcohol based hand sanitizer.

Production of rhamnolipids nano-miselles is very brief need more details about the production of rhamnolipids.

Also add UPLC LC/ESI-MS method and procedure for the identification

Why author used higher concentration 312 μg/ml when significant results obtained against virus infectivity at 20 and 78 μg/ml, compared to virus only and the blank solution sample. Justify?

Author only performed acute skin irritation study for 72 Hrs. What about the chronic/long term irritation test? They performed or not?

Author perform skin sensitization study?

Author performed biochemical analysis of ALT and AST. There is no correlation of these parameters with skin or eye irritation test?

Conclusion should be more describe and emphasize on skin irritation.

Author Response

Dear Prof. Dr. Nicholas Dixon,

Editor-in-Chief
Antibiotics

April 20th 2022

Re: Submission of revised paper Manuscript ID: antibiotics-1666343 titled, “Rhamnolipids nano-micelles versus alcohol based-hand sanitizer: A comparative study for antibacterial activity against hospital acquired infections and toxicity concerns”

Dear Professor, Nicholas Dixon,

We would like to thank you for your email dated March 28th 2022 enclosing the reviewers’ comments. We have carefully reviewed the comments and have revised the manuscript accordingly. Our responses to reviewer 1, reviewer 2, and reviewer 3 are given in a point-by-point manner below. Changes in the manuscript are marked in yellow color.

English language and style

( ) Extensive editing of English language and style required
( ) Moderate English changes required
(x) English language and style are fine/minor spell check required
( ) I don't feel qualified to judge about the English language and style

Yes

Can be improved

Must be improved

Not applicable

Does the introduction provide sufficient background and include all relevant references?

( )

(x)

( )

( )

Are all the cited references relevant to the research?

(x)

( )

( )

( )

Is the research design appropriate?

( )

(x)

( )

( )

Are the methods adequately described?

( )

(x)

( )

( )

Are the results clearly presented?

(x)

( )

( )

( )

Are the conclusions supported by the results?

( )

(x)

( )

( )

Comments and Suggestions for Authors

The current manuscript is focused on rhamnolipids nano-micelles solution demonstrated a significant decrease of SARS-CoV2 virus infectivity compared to virus only and the blank sample. Moreover, acute irritation test revealed that rhamnolipids nano-micelles are biocompatible with skin after topical application and proved to have no toxic effect on eye tissues after being instilled into the eye.

Comments

Reviewer 2, Comment 1:

More literature should be added about rhamnolipids more effective over alcohol-based hand sanitizer.

Response to reviewer 2, Comment 1

Thanks for reviewer’s comments, however, we are the first authors who run a comparative study between rhamnolipids nano-micelles and alcohol-based hand sanitizer and we have referenced them in the manuscript and they are reference number 6 and 23

Reviewer 2, Comment 2:

Production of rhamnolipids nano-micelles is very brief need more details about the production of rhamnolipids.

Response to reviewer 2, Comment 2

We have written all details about production of rhamnolipids nano-micelles in our previous publication (references Number 6) that was published in Antibiotics Journal as well and we have cited it in the current manuscript – So, the reader should be redirected to our previous publication. However, we rewritten this part to clarify this information – kindly check methodology section, section 4.2.2

Reviewer 2, Comment 3:

Also add UPLC LC/ESI-MS method and procedure for the identification

Response to reviewer 2, Comment 3

We have written all details about UPLC LC/ESI-MS method and procedure for the identification in our previous publication that was published in Antibiotics Journal as well and we have cited it in the current manuscript (reference number 6) – So, the reader should be redirected to our previous publication. However, we rewritten this part to clarify this information – kindly check methodology section, section 4.2.1

Reviewer 2, Comment 4:

Why author used higher concentration 312 μg/ml when significant results obtained against virus infectivity at 20 and 78 μg/ml, compared to virus only and the blank solution sample. Justify?

Response to reviewer 2, Comment 4

We have responded to this in first reviewer’s comment 1, however, we have copied our response below for reviewer’s convenience;

As we are aiming to use rhamnolipids nano-micelles as a replacement for alcohol-based hand sanitizers in the COVID-19 pandemic, we are interested in using rhamnolipids nano-micelles solution with a concentration that is sufficient to eradicate both bacteria and SARS-CoV2. In our previous publications, we figured out the concentration that is effective to inhibit the growth of selected resistant bacteria that are commonly known to cause hospital acquired infections in Egyptian hospitals and we have reported MIC value to be 312 µg/mL. Thus, we have to use this concentration to eradicate these bacteria, thus, we investigated 312 µg/mL to assure that the rhamnolipids nano-micelles is sufficient to effectively eradicate SARS-CoV2 and resistant bacteria and at the same time compatible to skin. However, we have investigated lower concentrations of rhamnolipids nano-micelles to figure out if nano-micelles can eradicate the virus at lower concentration. SARS-CoV-2 infectivity was reduced by 93%, 99.2% when they exposed to rhamnolipids nano-micelles concentration equivalent to 20, and 78 µg/mL. Thus, there is still a possibility for the virus to spread among patients and healthcare workers. Therefore, it is safer to use 312 µg/mL to assure both antiviral, and antibacterial activity.  Kindly check Discussion section, line 26 to 31.

Reviewer 2, Comment 5:

Author only performed acute skin irritation study for 72 Hrs. What about the chronic/long term irritation test? They performed or not?

Response to reviewer 2, Comment 5

We did not perform long term irritation test; however, we would like to thank the reviewers for his comment and we will run the chronic irritation test and publish it as a fourth article concerning rhamnolipids nano-micelles and this will be also involving long term stability of the rhamnolipids nano-micelles, and we might run a pilot study on volunteers to investigate the hand sanitation effect of rhamnolipids nano-micelles versus alcohol-based sanitizers.

Reviewer 2, Comment 6:

Author perform skin sensitization study?

Response to reviewer 2, Comment 6

Thanks for reviewer’s comment, we have run the skin sensitization test and we have added it to the manuscript, kindly check results section number 2.4.3. and Methodology section number 4.2.5.4.

Reviewer 2, Comment 7:

Author performed biochemical analysis of ALT and AST. There is no correlation of these parameters with skin or eye irritation test?

Response to reviewer 2, Comment 7

Both AST and ALT are enzymes that expresses the general condition of liver in acute general cases. If there is any leakage of the Rhamnolipids nano-micelles into blood through abraded skin and it has the potential to harm the internal organs in the animal, the first organ to be affected is liver as it is responsible for detoxification, so, these enzyme were measured to investigate any abnormality in liver that might occur.

Reviewer 2, Comment 8:

Conclusion should be more describe and emphasize on skin irritation.

Response to reviewer 2, Comment 8

The conclusion section has been rewritten according to reviewers’ comment

We do appreciate reviewers’ comments and hope that we managed to answer each point clearly

Round 2

Reviewer 2 Report

Authors have revised the manuscript accordingly.